Serum proteomic profiling during the periovulatory period identifies preliminary candidate biomarkers of oocyte maturation in deslorelin-induced ovulation in dogs

Udomthanaisit Larindhorn 1 2
Roytrakul Sittiruk sittiruk@biotec.or.th 3
Kallayanathum Wirakan 1 2
Charoenlappanit Sawanya 3
Tharasanit Theerawat Theerawat.T@chula.ac.th t.theerawat1974@gmail.com 1 2
1 Department of Obstetrics, Gynaecology and Reproductions, Faculty of Veterinary Science, Chulalongkorn University , Bangkok , Thailand
2 Center of Excellence for Veterinary Clinical Stem Cells and Bioengineering, Department of Obstetrics, Gynaecology and Reproductions, Faculty of Veterinary Science, Chulalongkorn University , Bangkok , Thailand
3 Functional Proteomics Technology Laboratory, National Center for Genetic Engineering and Biotechnology, National Science and Technology Development Agency , Pathumthani , Thailand
Brygadyrenko Viktor
Electronic publication date: 2025 Oct 15
Publication date: 2025
Volume: 13
Electronic Location ID: e20106
Received 2025 Mar 6; Accepted 2025 Aug 28
Copyright: ©2025 Udomthanaisit et al.
Copyright year: 2025
Copyright holder: Udomthanaisit et al.
License: This is an open access article distributed under the terms of the Creative Commons Attribution License, which permits unrestricted use, distribution, reproduction and adaptation in any medium and for any purpose provided that it is properly attributed. For attribution, the original author(s), title, publication source (PeerJ) and either DOI or URL of the article must be cited.
License URL: https://creativecommons.org/licenses/by/4.0/

Keywords: Canine, Deslorelin, Oocyte maturation, Ovulation, Proteomics, Serum

Funding: 90th Anniversary of Chulalongkorn University Fund (Ratchadaphiseksomphot Endowment Fund) European Union’s Horizon 2020 Research and Innovation Program under the Marie Skłodowska-Curie action project “WhyNotDry” No. GA-101131087 Excellence Center for Veterinary Clinical Stem Cells and Bioengineering This study was funded by the 90th Anniversary of Chulalongkorn University Fund (Ratchadaphiseksomphot Endowment Fund), the European Union’s Horizon 2020 Research and Innovation Program under the Marie Skłodowska-Curie action project “WhyNotDry” (No. GA-101131087), and Excellence Center for Veterinary Clinical Stem Cells and Bioengineering. The funders had no role in study design, data collection and analysis, decision to publish, or preparation of the manuscript.

==============================
Background

The reproductive physiology of canines is unique from other mammals because oocyte maturation occurs about 48–72 hours after ovulation. This study aimed to evaluate the blood serum protein profile in canines during the periovulatory period by using shotgun proteomics to identify potential biomarkers of oocyte maturation.

Method

Anestrus female dogs (n = 9) were implanted subcutaneously with 4.7 mg of deslorelin to induce estrus and ovulation. After implantation, ovariectomy was performed based on the level of progesterone and vaginal cytology evaluations conducted every 48 hours. Simultaneously, serum samples were collected for proteomic analysis. The oocytes were flushed from the oviduct, and the oocyte maturation stage was identified. Based on oocyte staging, all samples were categorized into three groups (n = 3 dogs per group): pre-ovulation, ovulation with immature oocytes, and ovulation with mature oocytes. All serum samples were analyzed in triplicate (27 independent injections) using liquid chromatography-tandem mass spectrometry to investigate the protein profile.

Results

Proteomics analysis showed 11 proteins upregulated from three different groups: tubulin-specific chaperone D (TBCD); coiled-coil domain-containing protein 93 (CCDC93); WDFY family member 4 (WDFY4); calcium and integrin-binding protein 1 (CIB1); IQ motif containing E (IQCE); large ribosomal subunit protein uL23 N-terminal domain-containing protein (RPL23A); neuraminidase 4 (NEU4); G protein-coupled receptor kinase (GRK3); NF-keppaB inhibitor delta (NFKBID); leucine rich repeat containing 4B (LRRC4B); and Rho family-interacting cell polarization regulator 2 (RIPOR2). Among these proteins, NFKDIB, which are oocyte maturation markers in other mammalian species, was upregulated in the ovulation with mature oocyte group (P < 0.01). Therefore, NFKBID is a possible to be an oocyte maturation marker in canines, but further studies on larger populations are needed to confirm its potential.

Introduction

Domestic dogs have non-seasonal monoestrus, typically experiencing one or two cycles per year. Puberty in most breeds occurs between 6 and 14 months of age, which is directly related to breed size (Concannon, 2011). Unlike other mammals, the maturation of dog oocytes resumes after they are released from the follicles. This maturation process reveals two major uncommon aspects. First, the oocyte ovulates at the immature germinal vesicle (GV) stage and completes meiosis during tubal transportation (Toshihiko & Toshiimitsu, 1975). Second, the oocyte remains viable for several days in the uterine tube (Lopes et al., 2007). When the oocyte resumes meiosis the Meiosis I is completed in 48 h, while Meiosis II (MII) are observed 48–72 h after ovulation (Reynaud et al., 2006; Songsasen & Wildt, 2007). The ovulation time is important for canine reproductive biotechnology, such as breeding management, cloning, intracytoplasmic sperm injection, and artificial insemination, particularly when frozen–thawed semen is used (Reynaud et al., 2020).

Ideally, the luteinizing hormone (LH) surge is the most reliable variable for estimating the ovulation time, but it has limited use in clinical applications (Mason, 2018). The time between low half-life LH surge and ovulation is variable depending on the number of times the LH is measured per day and the method used to determine ovulation (Hase et al., 2000). Clinically, serum progesterone (P4) levels increase immediately before or at the start of the LH surge and are utilized as an indirect marker due to its ease of measurement by immunoassay (Conley et al., 2023; Mason, 2018). However, the P4 concentration is variable, depending on the assay, particularly within the critical concentration ranges 1–2 ng/mL (Gloria et al., 2018; Hollinshead & Hanlon, 2019). A higher probability of mature oocytes was observed at serum P4 levels of 6–15 ng/mL, as measured by electric chemiluminescence immunoassay (Lee et al., 2017a). This variability has significant implications for clinical decisions regarding insemination timing and gestation length, potentially impacting outcomes (Conley et al., 2023).

Deslorelin, a gonadotropin-releasing hormone agonist, has been used in canine breeding management for various purposes including the synchronization of estrus in female dogs to facilitate planned breeding programs and to ensure precise ovulation timing (Borges et al., 2015; Fontaine et al., 2011; Holumbiiovska et al., 2025; Kutzler, 2005; Walter et al., 2011). A previous study found deslorelin implants in female dogs resulted in pregnancy rates of 63.3% for induced estrus, compared to 66.7% for natural estrus cycles. This similar pregnancy rate indicated that deslorelin does not significantly disrupt the normal reproductive physiology in female dogs (Borges et al., 2015; Walter et al., 2011).

In the field of canine reproductive proteomics, protein profiles from various components, including follicular fluid, oocytes, and oviductal exosomes, have been investigated. Comparative analysis of intrafollicular fluid proteins in canines, conducted before and after the LH surge, has revealed significant differences, with complement factor B identified as one of the key proteins (Fahiminiya et al., 2010). However, neither LH nor follicle-stimulating hormone initiate meiotic resumption in dogs. Furthermore, distinct protein profiles have been identified in oocytes between the diestrus and anestrus phases, with several proteins playing key roles in critical processes, such as cell cycle regulation, fertilization, transcriptional regulation, and signaling pathways that are essential for oocyte development and fertilization (Pereira et al., 2019). Recent proteomic studies of canine in vitro oviductal cell-derived extracellular vesicles have identified 398 proteins crucial for oocyte development (Lee, Lira-Albarrán & Saadeldin, 2021).

Despite these advancements, a serum biomarker for oocyte maturation in dogs remains elusive. This study aimed to investigate the abundant serum proteins across different oocyte maturation stages during the periovulatory period to identify potential serum biomarkers for mature oocytes.

Materials & Methods

Animal and ethical statement

This study was designed as a prospective cross-sectional design. Blood samples were collected from client-owned dogs presented as clinical cases at the Small Animal Hospital, Faculty of Veterinary Science, Chulalongkorn University, Thailand. The owners provided informed consent before their dogs were enrolled in the study. The experiment received ethical approval in accordance with the guidelines set by the Chulalongkorn University Institutional Animal Care and Use Committee, Thailand (Protocol No. 2131027). All procedures were performed in compliance with relevant guidelines and regulations, and adhered to the Animal Research: Reporting of In Vivo Experiments guidelines.

Nine female mixed-breed dogs, without reproductive disease, at a similar stage of anestrus (120–170 days since previous estrus), aged 3–5 years, weighing 7–15 kg, were confirmed to be in the anestrus phase based on vaginal cytology showing less than 10% of superficial cells and serum P4 levels below one ng/mL (Kustritz, 2020; Rijnberk & Kooistra, 2010). Dogs that did not meet these criteria or presented any reproductive system illness were excluded from the study. The dogs were divided into three groups, with three dogs per group, based on the criteria outlined in Table 1: pre-ovulation; ovulation with immature oocytes; and ovulation with mature oocytes. Following oocyte flushing, dogs with mixed stages of oocyte maturation were excluded from the study.

Table 1 The inclusive criteria for the division of three sample group.

Sample group	Vaginal cytology	Progesterone	Oocyte	
Pre-ovulation	Superficial cells > 80%	<5 ng/mL	N/A	
Ovulation with immature oocyte	Superficial cells > 80%	4–10 ng/mL	Oocyte with confined germinal vesicle	
Ovulation with mature oocyte	Superficial cells >80% with increasing intermediate and parabasal cells	5–15 ng/mL	Oocyte with extrusion of 1st polar body and metaphase II plate	
Notes.

All criteria were modified from Gloria et al. (2018), Kutzler et al. (2003), Lee et al. (2017a), Reckers et al. (2022), and Saint-Dizier et al. (2014).

Estrus induction

The induction of estrus involved the subcutaneous implantation of a 4.7 mg deslorelin implant (Suprelorin®, Virbac, New Zealand) in the right shoulder area selected, in accordance with the manufacturer’s leaflet, for its ease of access and minimal impact on mobility (Driancourt & Briggs, 2020; Gontier et al., 2022; Romagnoli et al., 2023). The day on which the deslorelin implant was administered was designated as day 0, and all subsequent time points were recorded as consecutive days from this reference point. Vaginal cytology was stained using Diff-Quick®, and P4 was assessed using radioimmunoassay (RIA) every 48 h. The implant usually induces estrus and ovulation 8–16 days after implantation (Fontaine et al., 2011). The implant was removed on the day of OVH.

P4 level measurement

P4 levels were measured using RIA based on tritium-labeled P4, produced in-house according to the method described by Bodhipaksha (1981). This technique is considered the gold standard for P4 detection (Gloria et al., 2018; Hollinshead & Hanlon, 2019; Skenandore et al., 2017). The intra-assay and inter-assay coefficients of variation were 6.27% and 11.63%, respectively. Briefly, 0.1 mL of the P4 antibody was introduced into the serum sample, followed by incubation at room temperature for 1 h. Subsequently, P4 labeled with tritium was introduced, and the mixture was incubated at 4 °C for 18 h. Following this incubation, 0.2 mL of activated charcoal was added, and the solution was centrifuged at 1500g for 15 min. The supernatant was collected, and the bound radioactive level was measured (Bodhipaksha, 1981).

Vaginal cytology examination

A sterile cotton swab was introduced at the point of dorsal commissure of the vulva and gently rotated to collect vaginal cells. The sample was rolled onto a glass slide, air dried, and fixed with methanol for 5 min (Leigh, Raji & Diakodue, 2013). All glass slides were stained with Diff-Quick® for evaluation and evaluated under light microscopy at 400X magnification. Approximately 200–300 epithelial cells per smear were counted (Johnston, Kustritz & Olson, 2001). The identification of exfoliated vaginal cells was based on the criteria established by Reckers et al. (2022). Individual cells were not measured directly; instead, most were assessed based on morphological features, and cell size was estimated using the scale bar after image capture. All cytological evaluations were performed by a single observer to minimize inter-observer variability (Reckers et al., 2022).

Briefly, cells with a diameter of less than 20 µm were classified as parabasal cells. For cells with a diameter greater than 20 µm, the presence of a cornification line was assessed, distinguishing between none or slightly (0–1 cornification line) and moderate to significant (two or more cornification lines). Cells exhibiting no or slight cornification with a cell area ≥ 79.5 µm2 were categorized as intermediate cells. Cells that did not fit these criteria underwent further examination of the nucleus. Cells with a well-defined nucleus were classified as superficial cells, while those lacking a nucleus or possessing a pyknotic nucleus were identified as squamous cells or anuclear superficial cells.

Animal anesthesia and post operative care

The anesthesia protocol for ovariohysterectomy (OVH) was monitored by licensed veterinarians to ensure patient safety, comfort, and effective pain management. Prior to anesthesia, dogs were fasted for 12 h and underwent a thorough physical examination and pre-anesthetic bloodwork to assess their health status. Sedation was achieved using acepromazine (0.03 mg/kg intramuscularly) and morphine (0.3 mg/kg intramuscularly) to reduce anxiety and provide analgesia. Anesthesia was induced with propofol (4–6 mg/kg intravenously) to effect, followed by endotracheal intubation to maintain the patient’s airway. Maintenance of anesthesia was accomplished using isoflurane (1.5–2.0%) delivered in 100% oxygen, with adjustments made to maintain a surgical plane of anesthesia. Throughout the procedure, vital parameters, including those of heart rate, respiratory rate, oxygen saturation, end-tidal CO2, blood pressure, and body temperature, were continuously monitored.

To prevent surgical site infections, cephalexin (25 mg/kg intravenously) was administered as prophylactic antibiotics and amoxicillin-clavulanic acid (20 mg/kg orally) was prescribed for seven postoperative days. Pain management was addressed by carprofen (four mg/kg subcutaneously) post operative. Dogs were closely monitored during recovery, and carprofen was continued at a dose of two mg/kg orally every 12 h for three days to ensure adequate pain control. Dogs were returned to their owners after complete recovery.

Blood and oocyte collection

OVH was performed based on serial evaluations of serum P4 levels and vaginal cytology, conducted every 48 h. Blood and reproductive organs from each dog were obtained on the day of OVH for each group. Whole blood (four mL) was collected into a plain tube, placed upright, and left undisturbed at room temperature for 15–30 min to allow clot formation. The tube was then centrifuged at 1,500 g for 10 min. The serum was then transferred to a 1.5-mL microcentrifuge tube and stored at −80 °C until proteomic analysis. The reproductive organs were examined for follicles, corpus hemorrhagicum (CH), and corpus luteum (CL). The oviducts were freed from the surrounding adipose tissue and flushed with sterile phosphate-buffered saline (PBS) to retrieve in vivo cumulus oocyte complexes. The oocytes were denuded and observed under a light microscope (Eclipse Ts1-FL; Nikon, Kanagawa, Japan). Subsequently, the denuded oocytes were incubated at 37 °C for 45 min in a microtubule-stabilizing solution consisting of 25% glycerol, 50 mM KCl, 0.5 mM MgCl2, 0.1 mM ethylenediaminetetraacetic acid, 0.1 mM ethylene glycol-bis(β-aminoethyl ether)-N,N,N’,N’-tetraacetic acid, 1 mM 2-mercaptoethanol, 50 mM imidazole, and 4% Triton X-100, following the protocol described by Simerly & Schatten (1993). After incubation, the oocytes were briefly washed with PBS-bovine serum albumin (BSA), fixed, and stored in 4% paraformaldehyde for further immunofluorescent staining for cell cytoskeleton and chromatin configuration analyses. A summary of sample collection is illustrated in Fig. 1.

Figure 1 Summary of the experimental design for sample collection under each condition.

Anestrus female dogs were implanted with deslorelin, and vaginal cytology along with progesterone levels was monitored every 48 h. Subsequently, the dogs were ovariectomized based on the inclusive criteria. On the day of the operation, serum was collected for proteomics, and oocyte staging was performed. Created in BioRender.

Ovarian morphology observation

The follicles, CH, and CL were classified based on their gross morphological characteristics, following the criteria described by Concannon (2011) and England (2012). The follicles were identified as fluid-filled structures on the ovarian surface. The CH appeared as reddish, soft, and collapsed structures filled with blood, indicative of recent ovulation. The CL was recognized as firm, yellowish bodies consistent with luteal development.

Oocyte staining and assessment of the oocyte maturation stage

To detect microtubules, the fixed oocytes were incubated for 1 h at 37 °C in dark with monoclonal anti-α-tubulin (T5168, 1:100, Sigma-Aldrich) and PBS plus 0.1% (v/v) BSA (A3311; Sigma-Aldrich) and 0.1% (v/v) Triton-X (CAS 9002-93-1, Sigma-Aldrich). Consequently, goat anti-mouse second antibody conjugated with tetramethylrhodamine isothiocyanate (TRITC) (T5393, 1:100, Sigma-Aldrich) and 0.1% BSA in PBS was used for secondary staining was applied and incubated for 1 h under the same conditions (37 °C in the dark). After washing twice with 0.1% (v/v) BSA in PBS, the oocytes were incubated in Alexa Fluor488 phalloidin (Molecular Probes; Invitrogen, Waltham, MA, USA) in 0.1% BSA and PBS at a diluted ratio of 1:50 for actin microfilament staining also at 37 °C in the dark. 4′,6-diamidino-2-phenylindole (S33025, Invitrogen) was used to stain chromatin under the same conditions (37 °C in the dark) (Thiangthientham et al., 2022; Thiangthientham et al., 2023). The stained oocytes were observed under a fluorescent microscope (BX51; Olympus, Tokyo, Japan). Immature oocytes exhibited dispersed microtubules without the formation of a meiotic spindle, with compacted chromatin enclosed within the ooplasm. In contrast, mature oocytes clearly displayed a barrel-shaped meiotic spindle at the metaphase plate, with chromatin aligned at its center and the polar body present in the perivitelline space.

Proteomic analysis

Total protein concentration in each serum sample was determined using the Lowry method, with BSA used as the protein standard. Equal amounts of protein were subjected to liquid chromatography-tandem mass spectrometry (LC-MS/MS) analysis. Quality assurance was maintained by injecting trypsin-digested BSA and blank samples after every 10th experimental run, confirming instrument stability and minimal contamination. Technical reproducibility was assessed through triplicate analyses of each biological sample, providing robust quantitative data.

Prior to analysis, the protein sample underwent reduction of disulfide bonds through incubation with 10 mM dithiothreitol in 10 mM ammonium bicarbonate for 1 h at ambient temperature. Subsequently, the samples underwent incubation with 100 mM iodoacetamide in 10 mM ammonium bicarbonate in the dark for 1 h at room temperature to alkylate cysteine residues in the proteins. Trypsin digestion of the protein samples was accomplished with sequencing grade trypsin (Promega) at a ratio of 1:20 (w/w); enzymatic digestion proceeded overnight (14–16 h) at room temperature. Enzymatic digestion was halted by the addition of 0.1% formic acid (FA). The tryptic peptide samples were prepared for injection into an Ultimate3000 Nano/Capillary LC System (Thermo Fisher Scientific) coupled with a Hybrid quadrupole Q-Tof impact II™ mass spectrometer (Bruker Daltonics) equipped with a Nano-captive spray ion source. Digested peptides (one µL) were enriched on a µ-precolumn (300 µm i.d. X five mm) C18 Pepmap 100, five µm, 100 A (Thermo Fisher Scientific), separated on a 75 µm I.D. ×15 cm column packed with Acclaim PepMap RSLC C18, 3 µm, 100Å, nanoViper (Thermo Scientific, UK). The C18 column was enclosed in a thermostatted column oven set to 40 °C. Solvents A and B were 0.1% FA in water and 0.1% FA in 80% acetonitrile, respectively, and supplied on the analytical column. A gradient of 5–55% solvent B was used to elute the peptides at a constant flow rate of 0.30 µL/min for 30 minutes. Electrospray ionization was carried out at 1.6 kV by using CaptiveSpray. Nitrogen at a flow rate of 50 L/h was used as a drying gas. Collision-induced dissociation product ion mass spectra were obtained using nitrogen gas as the collision gas. Mass spectra (MS) and MS/MS spectra were obtained in positive-ion mode at 2 Hz over the range of 150–2,200 m/z. The collision energy was adjusted to 10 eV as a function of the m/z value. Liquid chromatograph mass spectrometry (LC-MS) analysis of each sample was performed in triplicate. The MS/MS raw data and analysis are deposited in the ProteomeXchange Consortium (http://proteomecentral.proteomexchange.org) via the jPOST partner repository (https://jpostdb.org) with the data set identifier PXD056339.

Bioinformatics and data analysis

The LC-MS/MS data underwent analysis with MaxQuant software (version 2.2.0.0) and the resultant proteins were searched in the UniProtKB/Swiss-Prot database against Canis lupus familiaris database for protein identification and to explore their associated biological processes, cellular components, and molecular functions (Tyanova, Temu & Cox, 2016). The analysis of proteomic data is commonly used with the web-based platform MetaboAnalyst version 6.0 (http://www.metaboanalyst.ca) as done in this study. The data were log transformed to produce a more symmetrical distribution, approximating a normal or near-normal distribution (Pang et al., 2024). Analysis of variance (ANOVA) was conducted to identify proteins exhibiting statistically significant differences (p < 0.05) within the datasets. Additionally, protein–protein interaction analysis was carried out using the STITCH database 5.0 for data validation (Szklarczyk et al., 2016).

Results

All nine dogs exhibited the first signs of proestrus within 10 days after implantation, achieving a 100% estrus rate in 6–10 days (mean ± SD 7.6 ± 1.5 days). The success rate of ovulation following implantation in groups that were ovariectomized after presumptive ovulation, as determined by the presence of the CL, was 100%. Ovulation occurred 7 and 21 days after implantation, as determined by vaginal cytology and serum P4 levels (Table 2). P4 levels ranged from 0.89–5.74 ng/mL (2.9 ± 2.5) prior to ovulation, 4.59–12.52 ng/mL (7.6 ± 4.3) during ovulation with immature oocytes, and 5.92 - 12.3 ng/mL (9.4 ± 3.2) in the group with mature oocytes. The mature oocytes demonstrated the presence of a meiotic spindle and first polar body (Fig. 2)

Table 2 The descriptive data of each dog after deslorelin implantation.

Dog No.	Age
(years)	Weight
(kg)	Implant to proestrus (days)	Implant to OVH* (days)	Progesterone level** (ng/mL)	Percentage of vaginal cytology on day of OVH	No. of CL***	No. of oocytes (%)	Oocyte stage	Notes	
						Parabasal cells	Intermediate cells	Superficial cells	Anuclear superficial cells					
No.1	3	12.5	6	10	0.89	0	5	69	0	0	0	N/A	Multiple small follicle
(1–3 mm)	
No.2	5	15.0	6	9	2.01	0	3	33	64	0	0	N/A	
No.3	3	9.6	9	13	5.74	0	3	54	43	0	0	N/A	
No.4	5	11.3	7	14	12.52	0	0	22	78	15	4 (26.7%)	immature		
No.5	4	10.2	6	15	4.59	0	0	16	84	8	6 (75.0%)	immature		
No.6	4	7.6	8	14	5.55	5	0	25	70	7	6 (85.7%)	immature		
No.7	3	14.8	10	21	5.92	0	1	37	52	5	1 (20.0%)	mature		
No.8	3	12.2	8	19	12.3	2	8	18	72	6	2 (33.3%)	mature		
No.9	5	14.3	9	17	10	10	11	44	35	10	3 (30.0%)	mature		
Notes.

* OVH: ovariohysterectomy

** Progesterone level: progesterone level on day of OVH

*** CL: Corpus lutea

Figure 2 Observation of canine oocyte.

An immature oocyte under a light microscope (A) is compared to a mature oocyte that exhibits a visible perivitelline space and the presence of a first polar body (black arrow) (B). The second metaphase plate, along with the extruded DNA of the first polar body, was stained with 4′,6-diamidino-2-phenylindole (white arrows) (C). Alexa 488 phalloidin staining (D) was used to label the microfilaments, while monoclonal α-tubulin with tetramethylrhodamine isothiocyanate (E) was employed to visualize the microtubules. The combined staining results are presented in the merged image (F), scale bar = 50 µm.

The 12,781 proteins in each serum sample were analyzed by gel-digestion with LC-MS/MS and visualized in a heatmap with the trend of uniquely expressed protein clusters in each group (Fig. 3A). Partial least squares discrimination analysis (PLS-DA) was used to provide a comprehensive description of the data among the three following group. The two-dimensional PLS-DA plot demonstrates the precision of the protein cluster within each group, highlighting a clear separation between groups (Fig. 3B).

Figure 3 Heatmap and partial least squares-discriminant analysis.

Heat map of canine serum proteins (A) classified in the following groups: pre-ovulation (red), ovulation with immature oocyte (green), and ovulation with mature oocyte (blue). Each column indicates a sample, and each row indicates a serum protein exhibiting various expression levels. Protein intensity is represented by color, ranging from very low (deep blue) to extremely high (dark brown). The PLS-DA (B) of components one and two in two dimensions was conducted on all identified proteins. The individual samples are presented by colored points, with the surrounding colored regions representing the 95% confidence intervals.

According to ANOVA followed by Tukey’s test, 11 proteins were significantly upregulated. These proteins were Tubulin-specific chaperone D (TBCD), Coiled-coil domain-containing protein 93 (CCDC93), WDFY family member 4 (WDFY4), Calcium and integrin-binding protein 1 (CIB1), IQ motif containing E (IQCE), Large ribosomal subunit protein uL23 N-terminal domain-containing protein (RPL23A), Neuraminidase 4 (NEU4), G protein-coupled receptor kinase (GRK3), NF-keppaB inhibitor delta (NFKBID), Leucine rich repeat containing 4B (LRRC4B), and Rho family-interacting cell polarization regulator 2 (RIPOR2), (Supplementary data 1). Notably, these proteins exhibited distinct expression patterns across the three experimental groups. The proteins TBCD, CIB1, and LRRC4B were prominently upregulated in pre-ovulation group (Fig. 4). In the ovulated with immature oocyte group, WDFY4, RPL23A, and CCDC93 exhibited significant upregulation (Fig. 5). The group corresponding to ovulated oocytes with mature status exhibited upregulation of NEU4, GRK3, IQCE, RIPOR2, and NFKBID (Fig. 6).

Figure 4 Box plot of upregulated protein expressions in the pre-ovulation.

Tubulin-specific chaperone D (A), Calcium and integrin-binding protein 1 (B), and Leucine rich repeat containing 4B (C), were prominently upregulated. The distributions of differences are shown in a box plot. y-axis = intensity of mass spectrum; x-axis = three experimental groups.

Figure 5 Box plot of upregulated protein expressions in the ovulated with immature oocytes group.

WDFY family member 4 (A), large ribosomal subunit protein uL23 N-terminal domain-containing protein (B), and coiled-coil domain-containing protein 93 (C) were prominently upregulated. The distributions of differences are shown in a box plot. y-axis = intensity of mass spectrum; x-axis = three experimental groups.

Figure 6 Box plot of upregulated protein expressions in the ovulated with mature oocytes group.

Neuraminidase 4 (A), G protein-coupled receptor kinase (B), IQ motif containing E (C), Rho family-interacting cell polarization regulator 2 (D), and NF-kappaB inhibitor delta (E) were prominently upregulated. The distributions of differences are shown in a box plot. y-axis = intensity of mass spectrum; x-axis = three experimental groups.

The protein–protein and protein–chemical interaction networks in each group were determined using the STITCH database (version 5.0). Due to limited data from the Canis lupus familalis database, the Homo sapiens database was used. According to the protein–oocyte maturation molecule interaction network (Fig. 7), six proteins, ADRBK2 (synonym of GRK3), CIB1, RPL23A, LRRC4B, NFKBID, and TBCD, demonstrated a connection with oocyte maturation proteins or hormones associated with the estrus cycle directly or indirectly. Among these, LRRC4B exhibited a high confidence network with protein kinase (confidence score > 0.700) are presented as thick lines. Protein kinase was consequently connected with cyclin-dependent kinase (CDK) 1 and CDK2. CIB1 also showed a medium level connection with CDK2, with related proteins showing confidence scores exceeding 0.4. Additionally, NFKBID displayed a connection with CDK2. Furthermore, TBCD, RPL23A, and ADRBK2 (GRK3) displayed strong connections with magnesium adenosine triphosphate (mgATP), both directly and indirectly, through interactions with CDK proteins, and hormones associated with the estrous cycle. In contrast, CCDC93, IQCE, NEU4, and WDFY4 did not exhibit any associations with oocyte maturation molecules.

Figure 7 Network of protein–oocyte maturation molecule interactions analyzed by STITCH version 5.0.

Six proteins, adrenergic beta receptor kinase 2 (ADRBK2), Calcium integrin-binding protein 1 (CIB1), large ribosomal subunit protein uL23 (RPL23A), leucine rich repeat containing 4B (LRRC4B), NF-keppaB inhibitor delta (NFKBID), and tubulin-specific chaperone D (TBCD), showed correlation with oocyte maturation hormones, including estrogen, progesterone, and prostaglandin. Coiled-coil domain-containing protein 93 (CCDC93), IQ motif containing E (IQCE), neuraminidase 4 (NEU4), and WDFY family member 4 (WDFY4) showed no association with oocyte maturation molecules. The strength of the associations at the functional level was evaluated using edge confidence scores. The strong relationships with high edge confidence scores (>0.700) are presented as thick lines.

Discussion

This study highlights a proteomic analysis to compare protein profiles in canine serum during the periovulation period following estrus induction by deslorelin. A limitation of our study was the comparison between natural and hormonally induced estrous cycles. Consequently, the outcomes of our study should not be interpreted as directly representative of the physiological processes associated with ovulation and oocyte maturation during natural estrus. Further studies should be conducted to compare natural and induced estrus in dogs. Additionally, all animals included in this study were client-owned and presented as clinical cases, which limited the ability to standardize the environmental conditions, diet, and housing. While the potential influence of these confounding factors was considered during data interpretation. Moreover, the small sample size of three dogs per group limits the statistical power of the study, the findings should be regarded as preliminary and require confirmation in controlled experimental settings.

Our findings highlight a significant upregulation of NFKBID in serum samples associated with mature oocytes, aligning with previous in vitro study in cattle that identified this protein family as marker of oocyte maturation (Paciolla et al., 2011). Furthermore, the NFKB protein family is also associated with key regulatory factors of oocyte maturation in mammals, including the mitogen-activated protein kinase (MAPK) pathway and CDKs (Schulze-Osthoff et al., 1997; Shepel et al., 2013). For estrus synchronization, our results revealed 100% success in estrus induction, consistent with previous studies (Borges et al., 2015; Chotimanukul et al., 2023; Fontaine et al., 2011; Ponglowhapan et al., 2018; Von Heimendahl & Miller, 2012; Walter et al., 2011; Yodphet et al., 2025). More importantly, dogs implanted with deslorelin resulted in a similar pregnancy rate compared to natural estrus (63.3% vs. 66.7%, respectively) (Walter et al., 2011). Suggesting that the overall function of the reproductive organs is not significantly altered by deslorelin treatment. However, Walter et al. (2011) the same study noted a shorter proestrus duration and a reduced number of embryos or oocytes in the induced group compared to naturally cycling female dogs (Walter et al., 2011). Volkmann et al. (2006) reported an average of 6.8 CL per bitch following deslorelin implant. However, no actual oocytes were retrieved (Volkmann et al., 2006). In our study, we observed a slightly higher average CL count of 8.5 per bitch. Nevertheless, the number of oocytes recovered may have been underestimated due to technical losses during sample collection. Specifically, the oviduct was dissected into the infundibulum, ampulla, and isthmus for a separate secretion collection study prior to oocyte flushing, which may have contributed to partial loss of ovulated oocytes.

In veterinary practice, P4 is routinely employed as an indirect indicator to ascertain the timing of ovulation in dogs, with the common assumption that when P4 levels range 4–10 ng/mL at ovulation, oocytes will reach the mature stage (MII) within three days following ovulation (Hollinshead & Hanlon, 2019; Songsasen & Wildt, 2007). However, our study found the mature oocytes within a P4 concentration range of 5.92–12.3 ng/mL, consistent with the findings of Lee et al. (2017a) who observed a higher probability of mature oocytes with P4 levels of 6–15 ng/mL. The variability in P4 levels influencs practitioner decisions regarding artificial insemination timing, consequently impacting litter size (Hollinshead & Hanlon, 2019). Moreover, previous studies hypothesized that P4 influenced oocyte maturation in dogs. However, P4 supplementation for oocyte culture can marginally improve the maturation rate to approximately 30% (Qin et al., 2022). This indicates that P4 is not a direct factor that principally controls the process of oocyte maturation in vitro and cannot be used to identify the stage of oocyte maturation in vivo without validation (Pereira et al., 2019).

Protein identification represents a significant advancement in understanding canid reproductive physiology, even with challenging sample collection. Our study revealed several proteins in the serum of estrus female dogs, 11 of which exhibited upregulation during the periovulatory period. Based on the protein–oocyte maturation molecule interaction network, six proteins, ADRBK2 (GRK3), CIB1, RPL23A, LRRC4B, NFKBID, and TBCD, not only interacted with endocrine control of ovarian function (Concannon et al., 2018) but also with the oocyte maturation pathway. LRRC4B and CIB1 were significantly upregulated proteins involved with the meiosis resumption protein network in the pre-ovulation group. LRRC4B is potentially linked to oogenesin-1 and its associated genes and is specifically expressed in oocytes from primary to large antral follicles in mice (Buchanan & Gay, 1996; Minami et al., 2003). In contrast, the direct involvement of CIB1 in oocyte proliferation has not been reported; it is known to regulate the MAPK/extracellular signal-regulated kinases extracellular signal-regulated kinases (ERK) pathway, impacting cell survival in cancer cells (Leisner et al., 2013). Regarding protein interaction networks in immature ovulated oocyte groups, only RPL23A exhibited interactions with other ribosomal protein, large subunit (RPL) family proteins, although its relevance to reproduction has not been documented. PRL proteins are related to P53 or the tumor suppressor protein, which regulates cell stress responses, including ribosomal stress, usually inducing apoptosis, cell cycle arrest, or senescence (Chakraborty, Uechi & Kenmochi, 2011). This suggests that PRL has some function related to the ovulated GV stage of dog oocytes in regulating meiosis arrest at ovulation.

Of the five candidate proteins upregulated in mature oocyte groups, NFKBID stands out as a potential direct marker of oocyte maturation in dogs. NFKB is known as a transitional factor that regulates mammalian miosis resumption involving MAPK, cyclic monophosphates, and various factors (Schulze-Osthoff et al., 1997; Shepel et al., 2013). Furthermore, dog oviductal cells highly express MAPK1/3, SMAD2/3, and BMP15 during the estrus stage which are related to canine oocyte maturation (Lee et al., 2017b). NFKBID not only interacts with CDK1 and CDK2 but is also associated with the NFKB protein family, which serves as a marker for oocyte maturation in bovine in vitro studies (Paciolla et al., 2011).

Conclusions

This study identified key proteomic changes during the periovulatory period in dogs following deslorelin-induced ovulation. Proteomic analysis revealed 11 proteins upregulated at various stages of oocyte development. Among these, the protein NFKBID emerged as a potential biomarker for oocyte maturation, linked to the MAPK pathway and CDK. Additionally, the proteins LRRC4B and CIB1 were identified prior to ovulation and highlighted for their involvement in the resumption of meiosis. These findings offer new insights into the molecular mechanisms underlying canine oocyte maturation, advancing reproductive biotechnologies and improving breeding strategies in canines. However, this proteomic evaluation is an initial step. Further large-scale studies and functional validation in conspecifics are required to confirm the role of these proteins, particularly NFKBID, as reliable markers for canine oocyte maturation. In addition, comparative analyses with naturally cycling females are essential to determine whether the observed expression patterns reflect those occurring under physiological estrus conditions.

Supplemental Information

Supplemental Information 1 List of candidate proteins and Gene Ontology terms determined using UniProtKB/Swiss-Prot

Supplemental Information 2 The ARRIVE guidelines

We gratefully acknowledge Thitiporn Thongsima, Thitida Pakdeesanaeha, Sirichai Techarungchaikul, Sawita Santiviparat, and Nicha Hansinlawat for their valuable assistance in sample collection. We also thank all staff at the Functional Proteomics Technology Laboratory, National Center for Genetic Engineering and Biotechnology (BIOTEC), National Science and Technology Development Agency, for laboratory facility support.

Additional Information and Declarations

Competing Interests

Author Contributions

Animal Ethics

Data Availability

The authors declare there are no competing interests.

Larindhorn Udomthanaisit conceived and designed the experiments, performed the experiments, analyzed the data, prepared figures and/or tables, authored or reviewed drafts of the article, and approved the final draft.

Sittiruk Roytrakul analyzed the data, authored or reviewed drafts of the article, and approved the final draft.

Wirakan Kallayanathum performed the experiments, authored or reviewed drafts of the article, and approved the final draft.

Sawanya Charoenlappanit performed the experiments, authored or reviewed drafts of the article, and approved the final draft.

Theerawat Tharasanit conceived and designed the experiments, performed the experiments, prepared figures and/or tables, and approved the final draft.

The following information was supplied relating to ethical approvals (i.e., approving body and any reference numbers):

This experiment was ethically approved according to the guidelines established by the Chulalongkorn University Animal Care and Use Committee (Protocol No. 2131027).

The following information was supplied regarding data availability:

The proteomics data is available at ProteomeXchange: PXD056339.

The data is available in the Supplemental File.

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
