# Peer review of "Serum proteomic profiling during the periovulatory period identifies preliminary candidate biomarkers of oocyte maturation in deslorelin-induced ovulation in dogs"

_PeerJ, doi:10.7717/peerj.20106_

## Round 0.1 · original submission · Major Revisions

Dear Dr. Udomthanaisit,

Please take into account the reviewers' fundamental comments and improve the manuscript. I hope that the new version of this article will be approved by the reviewers.

Reviewer 1 ·

Basic reporting

Thank you for the opportunity to review the manuscript. Overall, I believe the paper is well-structured and presents valuable information. Given that it is a preliminary study, as the authors themselves acknowledge, it might be suitable for publication as a Short Communication.
However, there are several aspects that should be addressed prior to acceptance. First, it is important that the authors specify the kits used, for example, for progesterone measurement. I have also highlighted some sentences in the text that are unclear and require improved phrasing.
Regarding the language, I recommend a thorough revision of the English throughout the manuscript, as several sections are difficult to understand for an international audience. I suggest the authors seek assistance or consider using a professional editing service.
Finally, the references should follow a consistent format. For example, reference 1 is incomplete and reference 12 appears in all capital letters, which should be corrected.
Please feel free to contact me if you need any further clarification.

Experimental design

The experimental design is appropriate, as its limitations are clearly described. However, for future studies, it would be desirable to have detailed information on each bitch’s estrous cycle history. This would allow for a more accurate determination of the specific stage of anestrus, since early and late anestrus can present significant physiological differences.

Validity of the findings

The validity of the results appears to be sound. However, the manuscript does not indicate whether the assumptions for ANOVA were tested—specifically, whether the normal distribution of the data and the homogeneity of variances were verified.

Annotated reviews are not available for download in order to protect the identity of reviewers who chose to remain anonymous.

Reviewer 2 ·

Basic reporting

The manuscript is clear and use professional English, well-structured with relevant methodology.

Experimental design

The design is clear and easy to follow.

Validity of the findings

The findings is novel and valid.

Additional comments

Dear Editor,
The manuscript entitled “Serum proteomic profiling during the periovulatory period identified candidate biomarkers of oocyte maturation in deslorelin-induced ovulation in dogs” (#114838). The study investigated serum protein profiles in female dogs during the periovulatory period, aiming to identify potential biomarkers for oocyte maturation. Using a deslorelin-induced ovulation model in anestrus dogs (n=9), serum samples were collected and analyzed via shotgun proteomics (LC-MS/MS) alongside oocyte staging from oviductal flushes. Dogs were grouped based on oocyte maturation status: pre-ovulation, ovulation with immature oocytes, and ovulation with mature oocytes (n=3 dogs per group). The proteomic analysis revealed 11 upregulated proteins across the groups, with NFKB inhibitor delta (NFKBID)—a known oocyte maturation marker in other species—significantly upregulated in the group with mature oocytes (P < 0.01). These findings suggest NFKBID may serve as a potential biomarker for oocyte maturation in canines, although validation in larger populations is necessary.
In the field of small animal clinical reproduction, canine oocyte maturation exhibits distinctive characteristics that present unique challenges. To date, scientific literature specifically addressing serum biomarkers of oocyte maturation in the domestic dog (Canis lupus familiaris) is very limited. Existing research in canine reproductive biology has primarily focused on three areas: (i) hormonal profiling—such as LH and progesterone—to estimate ovulation timing; (ii) investigations of oocyte maturation utilizing follicular fluid analysis or in vitro culture systems rather than serum-based approaches; and (iii) studies on gene expression within oocytes and cumulus cells, with limited emphasis on circulating serum indicators. While the exploration of serum biomarkers for monitoring oocyte maturation in dogs is a promising and worthwhile direction for future research, the current manuscript in its present form, requires significant revision to meet publication standards.


Major concerns
1. The hypothesis and results presented in this manuscript is both novel and relevant, and the study offers valuable preliminary insights into the topic. However, the small sample size—three dogs per group—represents a limitation. To provide appropriate context and manage reader expectations, I recommend indicating that this is a preliminary study by including the phrase 'a preliminary study' in the title.
2. Although Walter et al. (2011) reported comparable pregnancy rates between deslorelin-induced and natural estrous cycles in bitches, other studies have yielded differing results. For instance, Ponglowhapan et al. (2018) documented variations in reproductive outcomes between natural and hormonally-induced cycles (Thai J Vet Med. 2018; 48(2): 211–217). Additionally, Chotimanukul et al. (2023) observed differences in serum concentrations and patterns of anti-Müllerian hormone (AMH)—a well-recognized marker of ovarian reserve and follicular activity—between natural and induced estrus in dogs (Animals (Basel). 2023; 13(2): 258). These findings, for instance, underscore that significant physiological differences may exist between natural and hormonally-induced estrous cycles in the bitch. Consequently, the outcomes of the present study should not be interpreted as directly representative of the physiological processes associated with ovulation and oocyte maturation during natural estrus. A discussion of these limitations and biological differences should be clearly included in the manuscript’s discussion section. Whether the findings observed in this study truly reflect those of natural estrus remains to be clarified in future investigations.

Minor concerns
1. Please check the formatting of the in-text citations to ensure they adhere to the journal’s reference style. For example, lines 50, 60, 167 etc.
2. Line 63 … However, the progesterone result is variable ….
3. In reference to the statement in Lines 64–66, what is the distribution or likelihood of finding mature versus immature oocytes in dogs numbered 4 to 9? Additionally, were all oocytes retrieved from each individual dog (as shown in Table 2) at the same maturation stage? The oocyte recovery rate reported in this study appears to be lower than those documented in previous publications. It is important that this discrepancy is addressed and discussed in the manuscript. Potential factors contributing to the lower recovery rate, such as the timing of collection, estrous cycle stage, stimulation protocol, or retrieval technique, should be considered to help contextualize the findings and assess methodological consistency with earlier reports.
4. Lines 71-77: any updated references?
5. Line 123: Please provide a more detailed description of the vaginal cytology examination. Specifically, clarify how the diameter of the cells was measured and indicate the number of cells evaluated. This information should be included in the manuscript to enhance the methodological transparency and reproducibility of the study.
6. Lines 159-160: How did you classify follicles, CH and CL in the canine ovaries? Any references? Please include in the manuscript.
7. Line 171: Images showing immature and mature oocytes should be included.
8. Lines 188-215: In proteomic analysis using MS and MS/MS, establishing the reliability and reproducibility of the technique is crucial. What are the controls and quality assurance practices in this study? This should be include in the manuscript.
9. The age and body weight of the animals used in the study should be included in the Results section to provide relevant background information.
10. Please carefully review the formatting of all references and ensure that in-text and reference citations are consistent with the journal’s guidelines. Specific examples that require attention include: Bodhipaksa (1981), Hase et al (2000), Littke and Kastelic (2006), and Tsutsui (1975).
11. The text in Figures 4, 5, and 6 is too small and difficult to read. Please enlarge the labels and annotations to improve clarity and readability.

Annotated reviews are not available for download in order to protect the identity of reviewers who chose to remain anonymous.

---

## Round 0.2 · Minor Revisions

Dear Dr. Udomthanaisit, I ask you to carefully correct all the shortcomings of the article and send me its final version, which can be approved by reviewers for publication.

Reviewer 2 ·

Basic reporting

Overall, I observe significant improvements in the revised manuscript, and all comments on the original version appear to have been thoroughly addressed.

Experimental design

no comment

Validity of the findings

no comment

Reviewer 3 ·

Basic reporting

Overall, the text is written in understandable English, but some parts are redundant and make it difficult to understand. In particular, the second paragraph of the Discussion section contains repeated descriptions of NFKBID and NFKB, repeated explanations of P4 and mature eggs, and multiple references to the involvement of MAPK and CDKs, making it difficult to read. The text needs to be revised to make it more concise.

There is no explanation for the white arrows a and b in the enlarged section of Figure 2F. It is unclear what the labels A to E on the right side of the graphs in Figures 4 to 6 indicate. It would be better to change the font type to match the gene names.

Experimental design

no comment

Validity of the findings

no comment

Reviewer 4 ·

Basic reporting

The manuscript presents an interesting and novel approach to identifying potential serum biomarkers of oocyte maturation in dogs. The identification of potential biomarkers associated with oocyte maturation, if they found the biomarker this would improving the timing of oocyte retrieval in canine it might be apply for use in clinical practice in the future.
However, the sample size is very small (n = 3 per group), which may make the results less reliable and harder to apply to larger population. Although the authors mentioned this, the limitation should be more clearly discussed.

Experimental design

- Please state clearly in the Materials and Methods section when the ovariectomy or ovariohysterectomy was performed in relation to implant administration and ovulation timing.
- In the abstract, the authors refer to the procedure as "ovariectomy," whereas in the Materials and Methods section, "ovariohysterectomy" is mentioned. Please clarify which surgical procedure was actually performed to ensure consistency throughout the manuscript.
- While the right shoulder is a common site for implantation, authors may consider briefly stating why this site was chosen (e.g., ease of access, consistency with previous studies).
- Please clarify the criteria used to determine the date for implant removal. Was this decision based on progesterone (P4) levels, vaginal cytology, or a combination of both?

Validity of the findings

the validity of the findings is limited by some factors:
-Small Sample Size: With only three dogs per group, the statistical power is low and the sample may not be representative of the broader canine population.
-Induced vs. Natural Estrus: Since estrus was hormonally induced with deslorelin, the physiological processes might differ from natural estrus cycles. This limits the direct applicability of the findings to normal reproductive physiology without further validation.
- protein function: some proteins were found to be increased during the periovulatory period, the study did not include follow-up experiments to confirm whether these proteins are truly involved in oocyte maturation.

Additional comments

- The authors should be consistency in the use of terminology "bitches" vs. "female dogs".
- Line 142: Please provide a reference supporting the use of cell size thresholds for vaginal cytology classification.
- Line 130: Consider rephrasing “gently rolled between the fingers” to something more scientific tone such as “gently rotated”.
- Oocyte staining: Please clarify How the staining steps were performed? at room temperature or at 4°C, and incubation was done in the dark (which is typical for fluorescent dyes). These are important for reproducibility.
- The authors mention trypsin digestion “overnight at room temperature.” Please specify approximate duration (e.g., 12–16 h).
- Line 237: please replace “againt” by “against”
- Line 348: please give the full term of PRL on first mention.
- Conclusion: for future studies, not only larger sample size but also consider on comparisons with natural estrus cycles and perform functional validation of the identified proteins. This would strengthen the biological relevance of your findings

---

## Round 0.3 · accepted · Accept

Dear Dr. Udomthanaisit, I congratulate you on the acceptance of this article for publication.

·

Basic reporting

The study is devoted to the search for markers of oocyte maturation in dogs based on the proteomic analysis of their blood serum. The authors conducted a fairly large and high-quality study worthy of publication. The manuscript is relevant, interesting for reproductive specialists and practicing veterinarians.
The text is written in understandable English and meets professional standards. The introduction includes the background and explains the general context of the problem to which the study is devoted. References to the literature are sufficient and correct to assess the relevance and coverage of the problem under study.
The structure of the article corresponds to the format of the journal.
The authors formulated a hypothesis and all the results are related to it.

Experimental design

The topic of the article corresponds to the purpose and problems of the journal, since it is devoted to the problem of animal reproduction, which is of interest to specialists not only veterinary specialists, but also biologists and physicians.
The methods are described in detail and contain information for reproduction.
In my opinion, the number of animals in the experimental groups is insufficient (n=3), so the results should be considered preliminary.

Validity of the findings

The authors have presented the results of the studies in a sufficiently high-quality manner.
The conclusions are clearly formulated, related to the original research question, and limited to confirmatory results.